# Inequalities in COVID-19 severe morbidity and mortality by country of birth in Sweden

Mikael Rostila ®[1,2] ✉, Agneta Cederström[1,2], Matthew Wallace ®[3], Siddartha Aradhya ®[3], Malin Ahrne[4] & Sol P. Juárez[1,2]

Migrants have been more affected by the COVID-19 pandemic. Whether this has varied over the course of the pandemic remains unknown. We examined how inequalities in intensive care unit (ICU) admission and death related to COVID-19 by country of birth have evolved over the course of the pandemic, while considering the contribution of social conditions and vaccination uptake. A population-based cohort study was conducted including adults living in Sweden between March 1, 2020 and June 1, 2022 (n = 7,870,441). Poisson regressions found that migrants from Africa, Middle East, Asia and European countries without EU28/EEA, UK and Switzerland had higher risk of COVID-19 mortality and ICU admission than Swedish-born. High risks of COVID-19 ICU admission was also found in migrants from South America. Inequalities were generally reduced through subsequent waves of the pandemic. In many migrant groups socioeconomic status and living conditions contributed to the disparities while vaccination campaigns were decisive when such became available.

The risk of COVID-19 infection, disease and death has not been equally distributed across ethnic, racial and migrant groups during the pandemic. A number of studies have found higher risks of COVID-19 infection, mortality and severe morbidity in ethnic minorities and migrants in Sweden[1–3] and elsewhere[4–11]. Most of the previous evidence has so far covered specific phases of the pandemic, predominantly the initial one[1,2]. Few studies have been able to provide evidence that encompasses—and differentiates between—major phases of the pandemic. This is relevant because the pandemic has been characterized by different waves, dominated by variants of the virus such as Delta or Omicron. COVID-19 infections, hospitalizations and deaths have varied markedly within and across waves[12,13].

It has been argued that migrants could be at higher risk of exposure to the virus due to adverse social and living environments[1,2,14]. Indeed, studies have shown that socioeconomic status and living conditions can at least partially account for the disparities in COVID-19 morbidity and mortality by country of birth[1,2,9,11,15]. In later waves, studies have recorded lower vaccination uptake among migrants[16]. While

vaccinations have been decisive for overall rates of severe illness and death, the contribution of the vaccination uptake in relation to the migrant-Swedish born inequalities remain unexplored.

Improving knowledge on the determinants contributing to COVID-19 disparities in infections, mortality and severe disease by country of birth is essential for designing preventive measures to reduce such disparities in future pandemics or health crisis. The aim of this paper is to examine how inequalities in hospitalization in intensive care unit (ICU) admission and death related to COVID-19 between migrants and Swedish born have evolved over the course of the pandemic, while also considering the contribution of vaccination uptake to these differences.

## Results

Table 1 shows descriptive information on the distribution of the adult population across region of origin, including the number, crude, and sex and age adjusted rates of COVID-19 ICU admissions and mortality. The crude rates for ICU admissions are highest for individuals born in

[1]Department of Public Health Sciences, Stockholm University, Stockholm, Sweden. [2]Centre for Health Equity Studies (CHESS), Stockholm University/ Karolinska Institutet, Stockholm, Sweden. [3]Demography Unit, Department of Sociology, Stockholm University, Stockholm, Sweden. [4]Department of Women's and Children's Health, Karolinska Institutet, Stockholm, Sweden. ✉e-mail: mikael.rostila@su.se

**Table 1 | Distribution of study population across region of origin, with number of individuals, number of COVID-19 related ICU admissions and mortality, with unadjusted, and age and sex adjusted rates**

| Region/country of birth | N(%) | COVID-19 mortality | COVID-19 mortality rate per 100,000 people | Age and sex standardized COVID-19 mortality rate per 100,000 people | COVID-19 ICU admission | COVID-19 ICU admission incidence rate per 100,000 people | Age and sex standardized COVID-19 ICU admission rate per 100,000 people |
|---|---|---|---|---|---|---|---|
| Sweden | 6,147,860 (78.2) | 15,063 | 110.5 (108.8–112.3) | 98.8 (97.2–100.4) | 5350 | 39.3 (38.2–40.3) | 37.5 (36.5–38.6) |
| Nordics w/o Sweden | 213,623 (2.7) | 1205 | 257.9 (243.4–272.5) | 144.5 (136.2–154.1) | 379 | 81.2 (73.0–89.4) | 56.9 (50.8–64.3) |
| EU28/EEA w/o Nordics | 327,230 (4.2) | 751 | 103.1 (95.7–110.5) | 121.6 (113.0–130.7) | 372 | 51.1 (45.9–56.3) | 60.8 (54.7–67.5) |
| Europe w/o EU28/EEA | 248,786 (3.2) | 603 | 108.7 (100.0–117.3) | 214.9 (197.0–234.2) | 616 | 111.1 (102.4–119.9) | 134.2 (123.1–146.6) |
| Middle East | 389,327 (5.0) | 590 | 67.6 (62.2–73.1) | 265.4 (241.8–291.1) | 1033 | 118.5 (111.3–125.8) | 177.8 (164.3–192.7) |
| Africa | 174,080 (2.2) | 193 | 49.4 (42.4–56.4) | 220.9 (182.4–268.5) | 358 | 91.8 (82.3–101.3) | 166.7 (143.7–197.5) |
| Asia | 253,705 (3.2) | 182 | 32.0 (27.3–36.6) | 193.2 (160.7–231.5) | 392 | 68.9 (62.1–75.7) | 157.2 (135.7–183.3) |
| North America | 24,832 (0.3) | 29 | 52.3 (33.3–71.4) | 68.4 (43.0–110.3) | 12 | 21.7 (9.4–33.9) | 27.5 (13.6–59.2) |
| South America | 77,713 (1.0) | 115 | 66.1 (54.0–78.1) | 167.1 (134.3–206.9) | 193 | 111.0 (95.4–126.7) | 137.3 (116.5–163.4) |

the Middle East (118.5), Europe excluding EU28/EEA (111.1) and South America (111.0), while the highest crude rates for COVID-19 mortality is found for individuals born in the Nordic countries (257.9) and Sweden (110.5). When adjusted for age and sex the rates for ICU admissions are still highest for individuals born in the Middle East (177.8), but individuals from Africa (166.7) and Asia (157.2) have the next highest, while the highest age and sex standardized rates for COVID-19 mortality is now also found for individuals born in the Middle East (265.4) and Africa (220.9). Table 2 shows the distribution of the adult population across sociodemographic variables including living conditions and vaccination status at the end of follow-up. Here one can see that many migrant groups have lower socioeconomic status, live in crowded apartments, in more densely populated neighborhoods, and have lower vaccination uptake than the Swedish born. Individuals from Africa and the Middle East have lower levels of education (31.1% and 26.2% have primary school education compared to Sweden 14.7%), lower incomes (42.1% and 37.4% in the lowest, 0–24 income quartile compared to Sweden 23.1%), and more often work in broad skill level 1-Elementary occupations (12.3% and 7% compared to Sweden 2.7%). The majority live in apartments (83.7% and 72% compared to Sweden 39.0%), in crowded conditions (59.3% and 50.1% in the lowest quartile of living area per person compared to Sweden 13.4%) and in densely populated neighborhoods (42.9% and 35.2% in the highest quintile compared to Sweden 18.3%). The proportionate differences are similar in many of the other migrant groups with some variations. In terms of vaccination uptake, these are lower in all migrant groups compared to Swedish born (90.9%) such as Africa (70.5%), Middle East (74.8%), and Europe without EU28/EEA (69.3%).

Figure 1 shows the relative risks of COVID-19 related ICU admission and mortality dependent on region of birth for the population across the entire risk period. All migrant groups except for individuals from North America have higher relative risk of ICU admission due to COVID-19 than the Swedish born population after adjustment for sex and age. Especially large excess risks in ICU admission can be seen for individuals born in Africa (RR = 4.1, 95% CI: 3.7, 4.6), the Middle East (RR = 4.3, 95% CI: 4.0, 4.6), Asia (RR = 3.7, 95% CI: 3.3, 4.1), South America (RR = 3.6, 95% CI: 3.1, 4.1) and Europe without EU28/EEA (RR = 3.5, 95% CI: 3.2, 3.8). These increased risks for ICU admission are partially explained by socioeconomic status, living conditions, as well as vaccination status, with relative risks attenuating by 54.8% for individuals born in Africa (RR = 2.5, 95% CI: 2.2, 2.8), by 43.7% in the Middle East (RR = 3.0, 95% CI: 2.7, 3.2), by 36.6% in Asia (RR = 2.7, 95% CI: 2.4, 3.0), and by 40.3% in South America (RR = 2.5, 95% CI: 2.2, 3.0).

Similar patterns are seen in COVID-19 related mortality, e.g., individuals born in Africa (RR = 3.1, 95% CI: 2.7, 3.6), the Middle East (RR = 2.7, 95% CI: 2.5, 2.9), Europe without EU28/EEA (RR = 2.3, 95% CI: 2.1, 2.5) and Asia (RR = 2.3, 95% CI: 2.0, 2.6) have substantially higher relative risk than Swedish born. The excess risks are reduced in the adjusted model by 64.6% for individuals born in Africa (RR = 1.8, 95% CI: 1.5, 2.0), by 63.9% in the Middle East (RR = 1.7, 95% CI: 1.5, 1.8), by 51.9% in Europe without EU28/EEA (RR = 1.7, 95% CI: 1.5, 1.8) and by 58.8% in Asia (RR = 1.6, 95% CI: 1.3, 1.8). Tables 1 and 2 in the Supplementary Information includes the complete set of results for all available models, and Supplementary Table 3 gives the percentages of excess risk explained by the four additionally adjusted models. It shows that the contribution of socioeconomic status, living conditions and vaccination status for COVID-19 ICU admission and mortality varies substantially between region of origin groups, explaining only a small amount of the higher ICU admission and mortality risk of the migrant groups from the Nordics and EU/EEA, alongside a much more substantial amount for migrants from the Middle East, Africa, Asia, South America, and Europe outside the EU/EEA.

In Fig. 2, we see how the relative risks for COVID-19 related ICU admission and mortality evolve over the four waves for models adjusted for age and sex and models additionally adjusted for

**Table 2 | Distribution of the study population across sociodemographic variables including SES, living conditions and vaccination status**

| | Sweden | Nordics w/o Sweden | EU28/EEA w/o Nordic | Europe w/o EU28/EEA | Middle East | Africa | Asia | North America | South America |
|---|---|---|---|---|---|---|---|---|---|
| **Sex** | | | | | | | | | |
| Female | 3,081,265 (50.1) | 123,227 (57.7) | 158,609 (48.5) | 127,638 (51.3) | 174,187 (44.7) | 82,699 (47.5) | 139,316 (54.9) | 12,058 (48.6) | 40,278 (51.8) |
| Male | 3,066,595 (49.9) | 90,396 (42.3) | 168,621 (51.5) | 121,148 (48.7) | 215,140 (55.3) | 91,381 (52.5) | 114,389 (45.1) | 12,774 (51.4) | 37,435 (48.2) |
| **Age category** | | | | | | | | | |
| 20–44 | 2,390,652 (38.9) | 34,689 (16.2) | 163,129 (49.9) | 120,555 (48.5) | 232,457 (59.7) | 118,129 (67.9) | 176,770 (69.7) | 13,409 (54.0) | 39,264 (50.5) |
| 45–64 | 1,983,418 (32.3) | 74,354 (34.8) | 96,190 (29.4) | 91,127 (36.6) | 126,325 (32.4) | 47,200 (27.1) | 65,689 (25.9) | 8022 (32.3) | 28,027 (36.1) |
| 65+ | 1,773,790 (28.9) | 104,580 (49.0) | 67,911 (20.8) | 37,104 (14.9) | 30,545 (7.8) | 8751 (5.0) | 11,246 (4.4) | 3401 (13.7) | 10,422 (13.4) |
| **Education** | | | | | | | | | |
| Primary | 903,943 (14.7) | 50,467 (23.6) | 31,231 (9.5) | 53,650 (21.6) | 101,883 (26.2) | 54,070 (31.1) | 48,720 (19.2) | 1162 (4.7) | 10,498 (13.5) |
| Secondary | 2,796,253 (45.5) | 86,934 (40.7) | 103,760 (31.7) | 100,441 (40.4) | 125,001 (32.1) | 65,499 (37.6) | 68,887 (27.2) | 4320 (17.4) | 30,556 (39.3) |
| Post-secondary | 2,418,167 (39.3) | 66,694 (31.2) | 151,207 (46.2) | 82,429 (33.1) | 145,092 (37.3) | 44,134 (25.4) | 112,537 (44.4) | 17,374 (70.0) | 34,370 (44.2) |
| Unknown | 29,497 (0.5) | 9528 (4.5) | 41,032 (12.5) | 12,266 (4.9) | 17,351 (4.5) | 10,377 (6.0) | 23,561 (9.3) | 1976 (8.0) | 2289 (2.9) |
| **Disposable income quartiles** | | | | | | | | | |
| 0–24 | 1,420,480 (23.1) | 80,979 (37.9) | 116,785 (35.7) | 69,258 (27.8) | 145,697 (37.4) | 73,374 (42.1) | 96,763 (38.1) | 8327 (33.5) | 24,317 (31.3) |
| 25–49 | 1,565,365 (25.5) | 56,242 (26.3) | 81,055 (24.8) | 61,684 (24.8) | 112,849 (29.0) | 48,958 (28.1) | 62,129 (24.5) | 4852 (19.5) | 19,629 (25.3) |
| 50–74 | 1,555,343 (25.3) | 41,815 (19.6) | 74,825 (22.9) | 73,473 (29.5) | 87,034 (22.4) | 36,598 (21.0) | 59,661 (23.5) | 5332 (21.5) | 19,979 (25.7) |
| 75–100 | 1,606,672 (26.1) | 34,587 (16.2) | 54,565 (16.7) | 44,371 (17.8) | 43,747 (11.2) | 15,150 (8.7) | 35,152 (13.9) | 6321 (25.5) | 13,788 (17.7) |
| **Broad skill level (ISCO-08 grouping)[a]** | | | | | | | | | |
| 3,4 | 1,930,890 (31.4) | 43,401 (20.3) | 79,689 (24.4) | 43,366 (17.4) | 56,508 (14.5) | 15,675 (9.0) | 50,270 (19.8) | 10,192 (41.0) | 18,142 (23.3) |
| 2 | 1,969,631 (32.0) | 46,187 (21.6) | 85,373 (26.1) | 88,428 (35.5) | 120,429 (30.9) | 67,555 (38.8) | 76,632 (30.2) | 4256 (17.1) | 27,945 (36.0) |
| 1 | 164,109 (2.7) | 5139 (2.4) | 22,397 (6.8) | 25,744 (10.3) | 27,115 (7.0) | 21,414 (12.3) | 29,430 (11.6) | 681 (2.7) | 7812 (10.1) |
| AF | 11,066 (0.2) | 68 (0.0) | 91 (0.0) | 44 (0.0) | 41 (0.0) | – (0.0) | 58 (0.0) | 14 (0.1) | 49 (0.1) |
| X | 2,072,164 (33.7) | 118,828 (55.6) | 139,680 (42.7) | 91,204 (36.7) | 185,234 (47.6) | 69,428 (39.9) | 97,315 (38.4) | 9689 (39.0) | 23,765 (30.6) |
| **Household** | | | | | | | | | |
| Cohabitating | 3,749,919 (61.0) | 114,059 (53.4) | 162,147 (49.6) | 137,695 (55.3) | 192,631 (49.5) | 62,619 (36.0) | 121,432 (47.9) | 14,309 (57.6) | 36,709 (47.2) |
| Single | 1,947,654 (31.7) | 80,137 (37.5) | 83,719 (25.6) | 57,484 (23.1) | 89,391 (23.0) | 52,683 (30.3) | 52,860 (20.8) | 5729 (23.1) | 22,793 (29.3) |
| Other | 450,287 (7.3) | 19,427 (9.1) | 81,364 (24.9) | 53,607 (21.5) | 107,305 (27.6) | 58,778 (33.8) | 79,413 (31.3) | 4794 (19.3) | 18,211 (23.4) |
| **Accommodation type** | | | | | | | | | |
| House | 3,411,114 (55.5) | 100,722 (47.1) | 131,002 (40.0) | 70,981 (28.5) | 84,085 (21.6) | 16,196 (9.3) | 68,559 (27.0) | 9697 (39.1) | 19,283 (24.8) |
| Apartment | 2,398,466 (39.0) | 98,145 (45.9) | 168,703 (51.6) | 165,771 (66.6) | 280,415 (72.0) | 145,649 (83.7) | 159,802 (63.0) | 12,940 (52.1) | 53,982 (69.5) |
| Special student | 57,569 (0.9) | 1813 (0.8) | 7299 (2.2) | 1925 (0.8) | 7061 (1.8) | 4704 (2.7) | 12,853 (5.1) | 816 (3.3) | 1086 (1.4) |
| Elderly care | 82,474 (1.3) | 5164 (2.4) | 2649 (0.8) | 1323 (0.5) | 1680 (0.4) | 805 (0.5) | 815 (0.3) | 169 (0.7) | 442 (0.6) |
| Other | 198,237 (3.2) | 7779 (3.6) | 17,577 (5.4) | 8786 (3.5) | 16,086 (4.1) | 6726 (3.9) | 11,676 (4.6) | 1210 (4.9) | 2920 (3.8) |
| **Living area per person quartiles** | | | | | | | | | |
| Q1 | 825,750 (13.4) | 24,200 (11.3) | 98,643 (30.1) | 94,368 (37.9) | 195,010 (50.1) | 103,244 (59.3) | 116,371 (45.9) | 6625 (26.7) | 26,746 (34.4) |
| Q2 | 1,349,095 (21.9) | 40,682 (19.0) | 75,358 (23.0) | 63,982 (25.7) | 84,696 (21.8) | 31,349 (18.0) | 60,833 (24.0) | 6499 (26.2) | 20,502 (26.4) |
| Q3 | 1,691,387 (27.5) | 59,813 (28.0) | 67,604 (20.7) | 47,137 (18.9) | 56,201 (14.4) | 19,882 (11.4) | 41,861 (16.5) | 5705 (23.0) | 16,416 (21.1) |
| Q4 | 2,106,999 (34.3) | 82,340 (38.5) | 69,863 (21.3) | 36,686 (14.7) | 42,427 (10.9) | 15,461 (8.9) | 25,571 (10.1) | 5034 (20.3) | 11,983 (15.4) |

**Table 2 (continued) | Distribution of the study population across sociodemographic variables including SES, living conditions and vaccination status**

| | Sweden | Nordics w/o Sweden | EU28/EEA w/o Nordic | Europe w/o EU28/EEA | Middle East | Africa | Asia | North America | South America |
|---|---|---|---|---|---|---|---|---|---|
| **DeSO population density quintiles** | | | | | | | | | |
| Q1 | 1,238,027 (20.1) | 37,465 (17.5) | 46,790 (14.3) | 8699 (3.5) | 8968 (2.3) | 4069 (2.3) | 14,808 (5.8) | 2773 (11.2) | 3936 (5.1) |
| Q2 | 1,236,667 (20.1) | 41,975 (19.6) | 47,579 (14.5) | 28,663 (11.5) | 43,912 (11.3) | 17,133 (9.8) | 31,109 (12.3) | 3091 (12.4) | 7297 (9.4) |
| Q3 | 1,312,207 (21.3) | 44,532 (20.8) | 59,055 (18.0) | 57,044 (22.9) | 83,700 (21.5) | 32,840 (18.9) | 50,222 (19.8) | 4265 (17.2) | 13,509 (17.4) |
| Q4 | 1,238,843 (20.2) | 45,549 (21.3) | 75,018 (22.9) | 68,376 (27.5) | 115,730 (29.7) | 45,358 (26.1) | 67,898 (26.8) | 5645 (22.7) | 19,992 (25.7) |
| Q5 | 1,122,116 (18.3) | 44,102 (20.6) | 98,788 (30.2) | 86,004 (34.6) | 137,017 (35.2) | 74,680 (42.9) | 89,668 (35.3) | 9058 (36.5) | 32,979 (42.4) |
| **Region** | | | | | | | | | |
| East | 1,648,797 (26.8) | 74,683 (35.0) | 131,259 (40.1) | 75,764 (30.5) | 139,210 (35.8) | 74,725 (42.9) | 103,194 (40.7) | 11,470 (46.2) | 42,058 (54.1) |
| Mid-west | 834,719 (13.6) | 16,923 (7.9) | 35,144 (10.7) | 33,283 (13.4) | 48,697 (12.5) | 16,311 (9.4) | 26,127 (10.3) | 1916 (7.7) | 5992 (7.7) |
| North | 807,678 (13.1) | 23,949 (11.2) | 14,695 (4.5) | 7767 (3.1) | 19,597 (5.0) | 18,727 (10.8) | 20,973 (8.3) | 1554 (6.3) | 4036 (5.2) |
| South | 882,572 (14.4) | 28,963 (13.6) | 64,829 (19.8) | 57,270 (23.0) | 66,087 (17.0) | 14,424 (8.3) | 37,324 (14.7) | 3961 (16.0) | 8566 (11.0) |
| South-East | 732,058 (11.9) | 29,803 (14.0) | 21,118 (6.5) | 17,823 (7.2) | 38,006 (9.8) | 19,514 (11.2) | 20,010 (7.9) | 1643 (6.6) | 4627 (6.0) |
| West | 1,242,036 (20.2) | 39,302 (18.4) | 60,185 (18.4) | 56,879 (22.9) | 77,730 (20.0) | 30,379 (18.2) | 46,077 (18.2) | 4288 (17.3) | 12,434 (16.0) |
| **Vaccinated** | | | | | | | | | |
| Yes | 5,589,732 (90.9) | 184,605 (86.4) | 210,576 (64.4) | 172,305 (69.3) | 291,253 (74.8) | 122,785 (70.5) | 208,948 (82.4) | 19,098 (76.9) | 63,238 (81.4) |
| No | 558,122 (9.1) | 29,018 (13.6) | 116,644 (35.6) | 76,472 (30.7) | 98,058 (25.2) | 51,282 (29.5) | 44,743 (17.6) | 5727 (23.1) | 14,472 (18.6) |

[a]3,4: Managers, professionals, technicians and associate professionals; 2: Clerical, service, and sales workers, skilled agricultural, forestry and fishery worker, craft and related trade workers, and plant and machine operators, and assemblers; 1: Elementary occupations; AF: Armed Forces; X: Not elsewhere classified.

socioeconomic status, living conditions, and vaccination status. In terms of COVID-19 related ICU admissions, the highest relative risks were observed in the first wave for most migrant groups. Individuals from Africa (RR = 8.2, 95% CI: 6.7, 10.2), the Middle East (RR = 6.2, 95% CI: 5.2, 7.4), and South America (RR = 5.8, 95% CI: 4.6, 7.3) had the greatest excess risks of ICU admission compared to those born in Sweden, risks which were attenuated after adjusting for socioeconomic and living conditions. However, as subsequent waves occurred, the relative risks gradually declined for most migrant groups and by the fourth wave no disparities in ICU admission rates were evident after adjusting for SES, living conditions, and vaccination status.

The pattern is somewhat different for COVID-19 mortality where for most migrant groups the disparities are highest in the first and third wave and the gradual decline across waves is not observed, although by the fourth wave there are no disparities left after adjusting for SES, living conditions, and vaccination status. Any excess risks in the second wave can be accounted for by SES and living conditions, since vaccinations had only begun to be administered at the end of the second wave.

In order to fully examine the role of vaccinations in the fourth wave, we compared the relative risks for COVID-19 related ICU admission and mortality in the fourth wave with and without adjusting for vaccinations. The results can be seen in Fig. 3. Most of the disparities could be accounted for by differential vaccine uptake in the country/region of origin groups. For example, the relative risks of ICU admission related to COVID-19 in individuals born in Europe excluding EU28/EEA reduces from 2.8 (1.9,3.9) to 1.5 (1.0,2.2) once accounting for differences in vaccination uptake. In some cases, such as for individuals born in EU28/EEA without the Nordics, Africa, Asia, and South America their RR are no longer significantly different from 1 compared to people born in Sweden when accounting for vaccination.

## Discussion

We found substantially higher risks of COVID-19 ICU admission and mortality among migrants from Middle-East, Africa, Asia, South-America and other European countries excluding EU28/EEA when compared to Swedish born individuals during the pandemic. These findings are in line with previous studies showing higher risks of COVID-19 ICU admissions and mortality in groups of migrants during specific phases of the pandemic[1–3]. The original contributions of our study are threefold. First, our follow-up (which ends in June 1, 2022) encompasses the entire pandemic to the point where COVID-19 was no longer considered a public health priority in Sweden. Second, we were able to demonstrate how disparities in COVID-19 related ICU admission and mortality by region/country of birth varied across the various waves of the pandemic. Third, we were able to explore how variation in vaccination uptake by region/country of birth influenced disparities in COVID-19.

We found that disparities in COVID-19 related ICU admission were profound in the first wave and tended to decrease during subsequent waves of the pandemic and were immaterial during the fourth wave after adjusting for SES, living conditions, and vaccination status. These findings indicate that migrants were more vulnerable to the virus during the initial stages of the pandemic when no vaccinations were available, the burden on the health care system was profound, and mitigation and treatment strategies were less developed. It has also been argued that public health messages and guidelines were insufficient for migrants during the initial phase of the pandemic[14] which could account for higher risk of infection.

Furthermore, high levels of exposure to COVID-19 in the migrant community during the first waves of the pandemic may have led to the deaths of many of the most vulnerable individuals and infected many others who developed immunity. The introduction of vaccinations in wave 3 could contribute to increasing inequalities in COVID-19 related

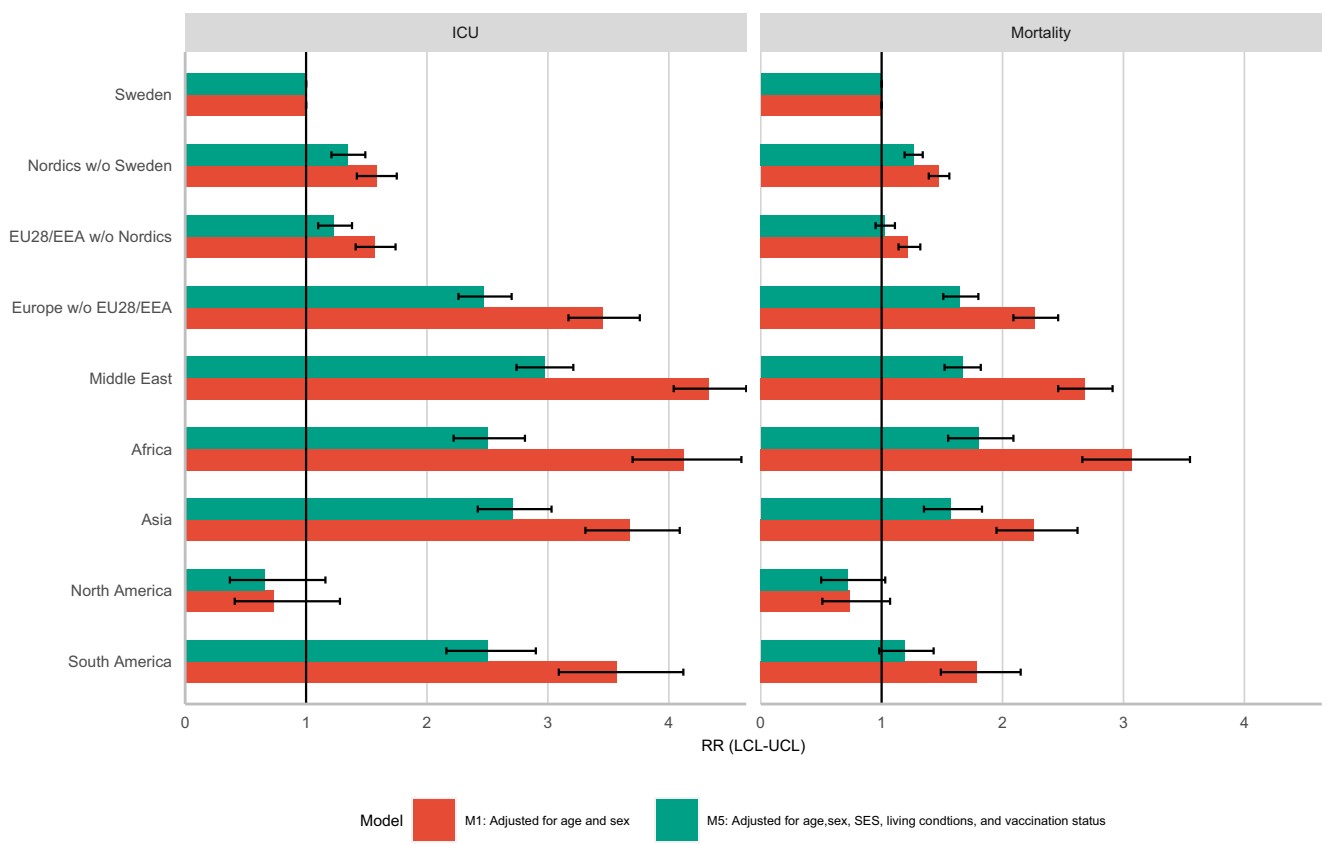

**Fig. 1 | Incidence rate ratios (RR) of COVID-19 related ICU admission and mortality by region/country of birth.** Mean estimates are derived from a study population of $n = 7{,}870{,}441$ with 18,731 COVID-19 related deaths and 8705 COVID-19 related ICU admissions. The error bars show the 95% confidence intervals.

mortality between migrants and Swedish born. The vaccination campaign reached the older population and certain risk groups first (overrepresented by Swedish born) while covering the middle-aged migrant population to a lesser extent. Also, restrictions on movement and social interactions decreased as vaccinations were administered to the most vulnerable groups which could lead to higher rates of infections and community spread of the virus. These circumstances could have contributed to a lower protection in migrants who were then more exposed to the virus. Another potential, although yet unexplored, mechanism is that migrants suffer discrimination that could influence health care access and quality of care and thereby contributes to a higher COVID-19 mortality and ICU. These inequalities decreased during the last wave when most of the population aged 18 or older had been offered a vaccine for COVID-19. The remaining differences between groups during the last wave could to a large extent be explained by differential vaccination uptake, which has been lower among migrant groups.

Our results show that differences in socio-economic status and living conditions contributed in part to the higher risk for COVID-19 related ICU admission and mortality in migrants, but other factors such as contact patterns could not be accounted for in our models. Socioeconomic conditions are strongly associated with several diseases and health risk behaviors that could increase the risk of COVID-19 death in migrants[10,17]. Thus, socioeconomic adjustments could account for some underlying co-morbidities, explaining the higher COVID-19 ICU admission and mortality in foreign-born[1]. Migrants are more likely to live in shared and overcrowded accommodations, neighborhoods with high population density and multigenerational housing with implications for increased infection rates in migrants and transmission of the virus between younger and older members of the household[1,17]. They are also disproportionately represented in front-line public-

facing jobs such as health care and the service industry that can place them at increased exposure of COVID-19 infection[1,13,17]. Healthcare seeking and barriers to care due to cultural and language barriers could also contribute to adverse COVID-19 outcomes in migrants[17] although a previous study indicated that language proficiency played a minor role for higher COVID-19 mortality in migrants[18].

We found that vaccination uptake did contribute to the disparities in COVID-19 mortality and morbidity by country of birth during the fourth wave, where vaccination status accounts for much or all of the excess risk. Our findings therefore indicate that preventive action focused on migrants social and living conditions could be important in future pandemics when no vaccine is yet available. Such action however requires longer-term investments and are difficult to achieve during an acute health crisis. When vaccinations do become available, providing equal access and use of vaccines quickly is the most effective tool to mitigate disparities in COVID-19 morbidity and mortality according to our findings. This is especially important when considering the lower vaccination uptake among migrants when compared to Swedish-born individuals[19].

### Strengths and limitations

The use of total population data, longitudinal follow-up covering the four waves of the pandemic, reliable information on COVID-19 ICU admission and mortality, vaccinations and socio-demographic information could be considered important strengths of this study. Some limitations of this study should be noted. The remaining excess risk of COVID-19 ICU admission and mortality found in foreign-born persons could indicate a higher prevalence of underlying adverse health conditions (e.g., diabetes, coronary health problems) and health risk behaviors (e.g., smoking, alcohol problems) found in the previous literature[20-24]. Although SES could be considered a proxy for some of

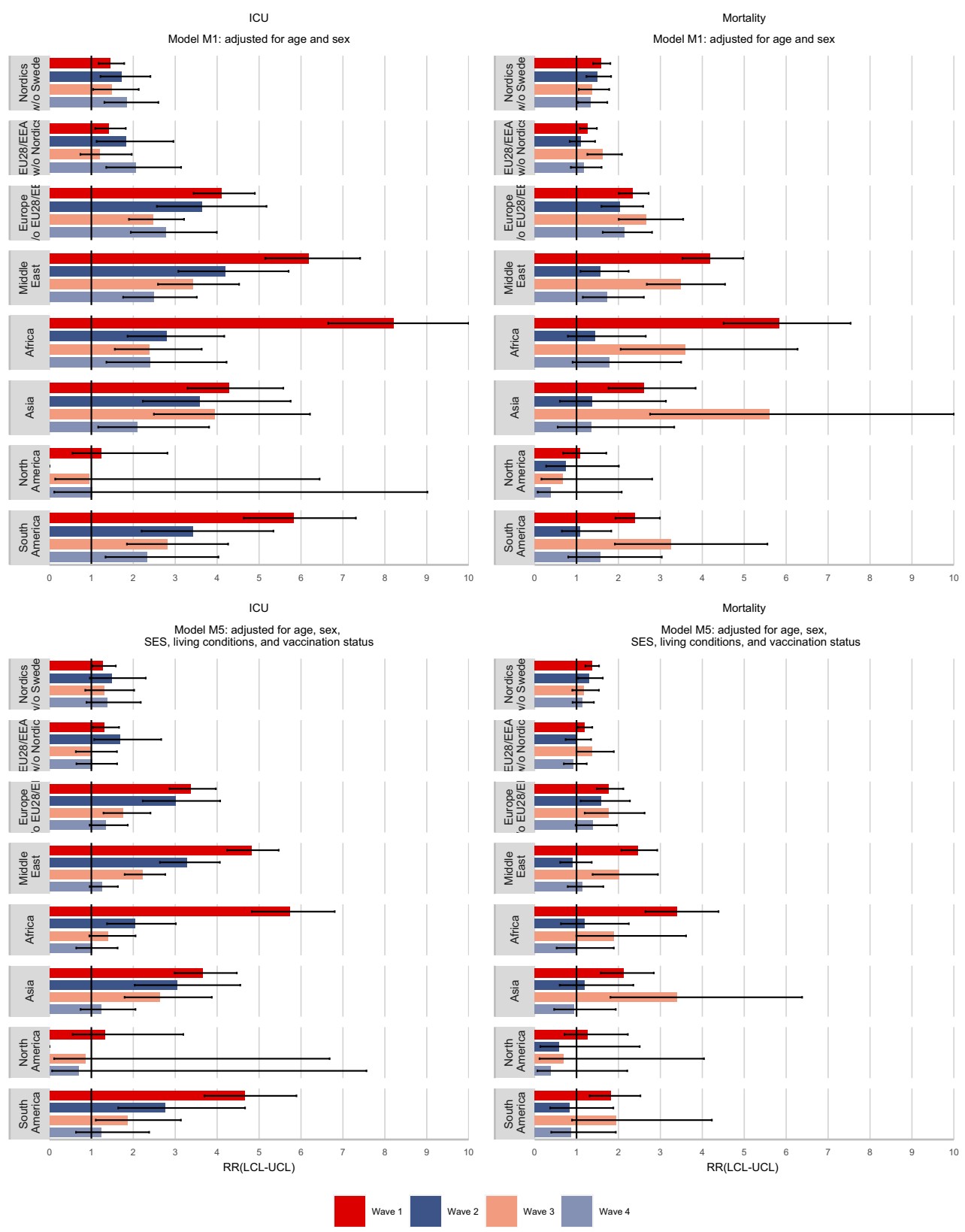

**Fig. 2 | Incidence rate ratios (RR) of COVID-19 related ICU admission and mortality by region/country of birth and wave of the pandemic.** Top panels show models adjusted for age and sex, and bottom panels show models additionally adjusted for SES, living conditions, and vaccination status. Mean estimates are derived from a study population of n = 7,870,441 with 18,731 COVID-19 related deaths and 8705 COVID-19 related ICU admissions. The error bars show the 95% confidence intervals.

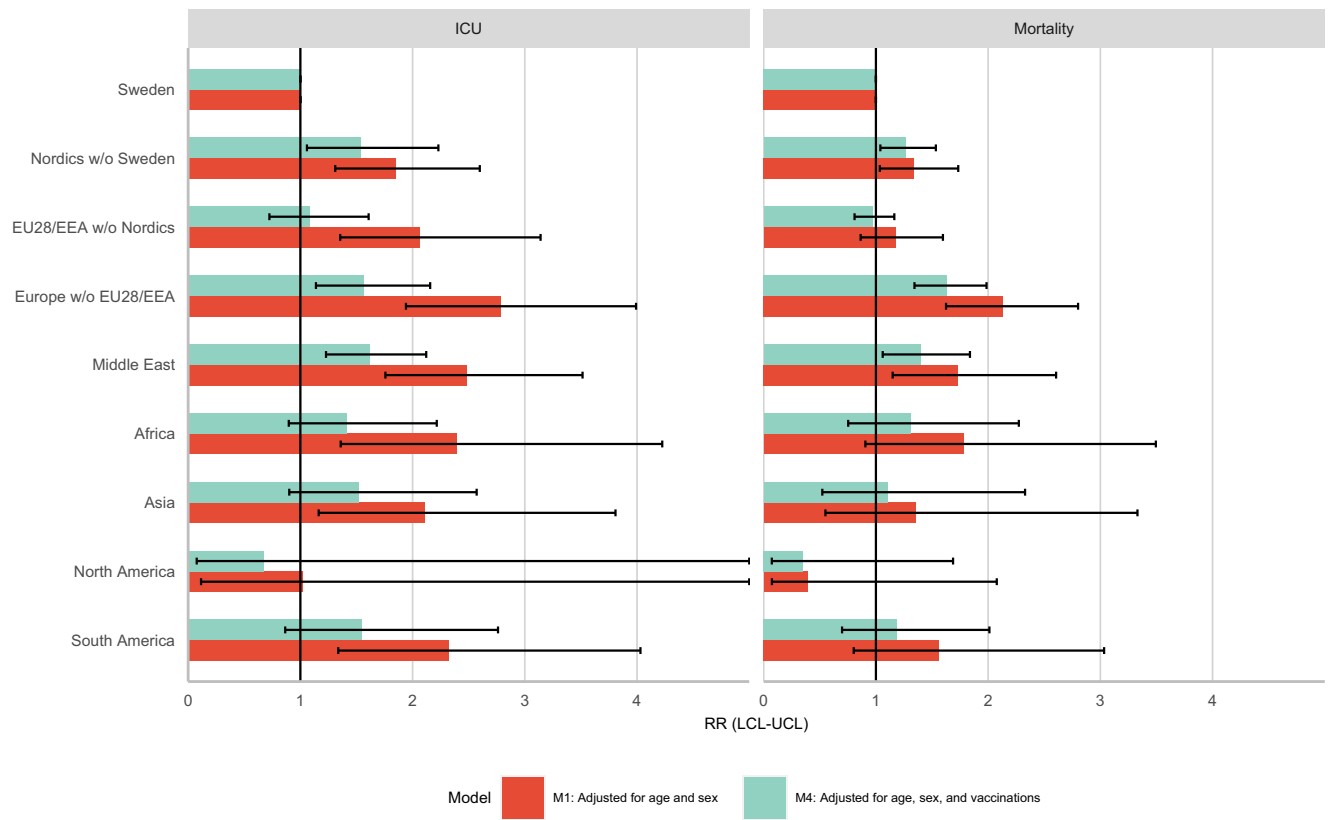

**Fig. 3 | Incidence rate ratios (RR) of COVID-19 related ICU admission and mortality by region/country of birth comparing with and without vaccination status during the fourth wave of the pandemic.** Mean estimates are derived from a study population of $n = 7,870,441$ with 18,731 COVID-19 related deaths and 8705 COVID-19 related ICU admissions. The error bars show the 95% confidence intervals.

these risk factors, we could not directly study underlying co-morbidities for COVID-19 and how they vary across migrant groups in our registry data. Additional information on social interaction patterns, language proficiency, adherence to guidelines and information, and cultural and religious attitudes and norms regarding virus transmission could contribute to the understanding of disparities in COVID-19 by country of birth. Unfortunately, our data did not include such information. The data also lack accurate information about infections to evaluate the contribution of differential exposure. Previous findings have also found that the incomplete ascertainment of COVID-19 related deaths contributes to an underestimation of the mortality risk[25]. This might impact overall COVID-19 mortality rates although it is unlikely that it explains the disparities in COVID-19 mortality by country of birth. Finally, ICU admission does not necessarily reflect disease severity and could depend on factors such as the availability and effectiveness of treatment. Therefore, there is a risk of confounding by time period and selection bias in our analyses although it seems less likely that these factors impact ICU admission disparities by country of birth.

Additionally, the use of traditional regression models such as Poisson have limitations in the modeling of infectious diseases[26] since events are not independent. However, the analyses are meant to be descriptive in terms of relative risks between groups and provide indications of the possible mechanisms. The estimates should not be interpreted as causal pathways since patterns of interaction and contact between individuals in different groups cannot be accounted for in the models. Moreover, the moderating effects of socioeconomic status and living conditions are not included in these models.

Much higher risks of COVID-19 related ICU admission and mortality were found in most migrant groups, especially in the initial phase, although disparities were reduced throughout the pandemic.

Disadvantaged socioeconomic status and living conditions partially explain the inequalities in severe COVID-19 disease and mortality by country of birth. The availability of a vaccine did play a decisive role when it became available. While preventive policies should focus on the social and living conditions of migrants in the absence of a vaccine, supporting equitable resourcing and access to vaccine at a scale and intensity according to the degree of needs of specific migrant populations would have been the most important preventive strategy in order to mitigate disparities on COVID-19 related health outcomes.

## Methods
This study complies with all relevant ethical regulations and was approved by the Swedish Ethical Review Authority (decision no 2021-05754-02 and no 2022-00428-01).

### Data
Data from multiple registers were linked using a pseudonymized personal identification number. The cohort was defined by the total adult population alive and residing in Sweden at the end of 2019. Information on background variables including country of birth were obtained from the Total Population Register, sociodemographic characteristics from the Longitudinal Integrated Database for Health Insurance and Labor Market Studies (LISA), living conditions from the Dwelling Units Register, intensive care and deaths in COVID-19 from the national notifiable disease registry (SmiNet), the Swedish Intensive Care Register, and the Cause of Death Register, and vaccinations against COVID-19 from the National Vaccination Register.

### Variables
Region/country of birth was categorized into born in Sweden, Nordics without Sweden, EU28/EEA without Nordics, Europe without EU28/

EEA, Middle East, Africa, Asia, North America, and South America. EU28/EEA includes the UK and Switzerland. Age was grouped into 20–29, 30–34, 35–39, 40–44, 45–49, 50–54, 55–59, 60–64, 65–69, 70–74, 75–79, 80–84, 85–89, 90–99, 100 and above. Education was derived from ISCED and was classified into Primary, Secondary, Post-Secondary, and Unknown, and individual disposable income was divided into quartiles. Broad skill level was derived from ISCO-08 (International Standard Classification of Occupations) and is categorized as the following, 3,4: Managers, professionals, technicians and associate professionals, 2: Clerical, service, and sales workers, skilled agricultural, forestry and fishery worker, craft and related trade workers, and plant and machine operators, and assemblers, 1: Elementary occupations, AF: Armed Forces, X: Not elsewhere classified. Household type was grouped into Cohabiting, Single, and Other. Since many COVID-19 deaths occurred in old-age facilities, the type of accommodation was included and defined by House, Apartment, Special/student accommodation, Elderly care, and Other accommodation. Living area per person was divided into quartiles and neighborhood (DeSO) population density was divided into quintiles. To account for the uneven spread of the virus in different parts of the country at different times, a regional indicator was also included. Administrative regions were grouped into six parts of the country as East, Mid-West, South, West, South-East, and North. Vaccination status was included as a binary yes or no variable and was based on receipt of the first vaccination only.

In total, 1131 individuals with unknown country of birth, 13,275 with missing DeSO neighborhood (73 individuals had both missing DeSO and unknown country of birth), and 83 with missing disposable income were excluded from the data set. Of the 14,416 individuals excluded, 48% were foreign born and 80% were in the lowest income quartile.

## Statistical analysis

We modeled mortality rates with confirmed COVID-19 or incidence rates of intensive care unit (ICU) admission with COVID-19 using Poisson regression analyses, and estimated incidence rate ratios dependent upon country/region of origin. For mortality, follow-up started from March 1 2020 and ended on June 1 2021. On and between these two dates, individuals could experience the event (i.e., death from COVID-19) or be censored due to (1) death from a cause other than COVID-19 or (2) reaching the end of the follow-up period alive. For ICU admission, follow-up started from March 1 2020 and ended on June 1 2021. On and between these two dates, individuals could experience the event (i.e., admission to ICU) or be censored due to (1) death from COVID-19 (having not been admitted to the ICU), (2) death from a cause other than COVID-19, or through reaching the end of the follow-up period alive having not been admitted to the ICU. The log of the follow-up was included as an offset in the models and incidence rate ratios (RR) were calculated with a 95% confidence interval (CI). Five models were run, the first minimally adjusted (M1) for sex and age, then another model adjusted for sex, age, and the sociodemographic factors education, income, occupation, and household type (M2). A third model included sex, age, and living conditions which include the housing and neighborhood characteristics, type of accommodation, living area per person, neighborhood population density and region of residence (M3). The fourth models adjusted for sex, age and vaccination status (M4), while the fifth model included all of the individual and residential level factors including vaccinations (M5). The Akaike Information Criterion (AIC) is calculated for each model in order to compare and test which of the models best fits the data. The follow-up was split on the date for the first vaccination if applicable, and vaccination was recorded as a time-varying variable. In order to evaluate how the risks evolved over the waves, the data was further split into seven calendar time periods, with breaks at 2020-03-01, 2020-07-07, 2020-10-25, 2021-02-01, 2021-06-01, 2021-12-01, 2022-05-01, 2022-06-01, and an interaction between region/country of birth and time period

was estimated. The first wave was between 2020-03-01 and 2020-07-07 (Wave 1), the second between 2020-10-25 and 2021-02-01 (Wave 2), and the third wave begins immediately after that and ends in 2021-06-01 (Wave 3). It is a matter of definition whether wave 2 and wave 3 are actually distinct waves given that neither ICU admissions or deaths actually decreased to the levels seen in the summer of 2020. However, because of the drastic decrease of COVID-19 related mortality in the winter and spring of 2021 and the fact that vaccinations had begun on a large scale, we chose to define these as separate waves. The fourth "omicron" wave is between 2021-12-01 and 2022-05-01 (Wave 4), which saw a sharp increase in confirmed cases while ICU admission and deaths related to COVID-19 were lower than in previous waves due to vaccinations and a milder variant of the virus. The waves are illustrated in Figs. 1 and 2 in the Supplementary Information for COVID-19 related ICU admission and mortality, and were derived by finding when mortality cases per day goes under 10, and ICU admission under 3. The cut-off date between wave 2 and 3 is when ICU admission reaches a local minimum. The statistical analyses were performed using R 4.2.2 (R Core Team, 2022; R Foundation for Statistical Computing, Vienna, Austria).

## Reporting summary

Further information on research design is available in the Nature Portfolio Reporting Summary linked to this article.

## Data availability

The data that support the findings of this study are available from The Swedish Public Health Agency but restrictions apply due to the sensitive nature of individual level health data. The research question needs ethical approval from the ethical review agency (Etikprövningsmyndigheten) and then a data request can be sent to Registerhantering@folkhalsomyndigheten.se. The timeframe for data access is up to 6 months, and data must be securely stored and handled in Sweden. The data are linked from the Total Population Register, the Longitudinal Integrated Database for Health Insurance and Labor Market Studies (LISA), the Dwelling Units Register, the national notifiable disease registry (SmiNet), the Swedish Intensive Care Register, the Cause of Death Register, and the National Vaccination Register. The registers are linked via a pseudonymized personal identification number. Data are however also available from the authors upon reasonable request and with permission of The Swedish Public Health Agency.

## Code availability

Custom R scripts produced in R 4.2.2 are available in Zenodo under a Creative Commons license[27].

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

## Acknowledgements

M.R., S.P.J., A.C. and S.A. were funded by the Swedish Research Council for Health, Working Life and Welfare (grant number 2021-00271). M.R., S.P.J. and A.C. were funded by the Swedish Research Council for Health, Working Life and Welfare (grant number 2016-07128). M.R. and A.C. were funded by a collaboration grant with the Swedish Public Health Agency.

## Author contributions

M.R., A.C., S.P.J., M.W., S.A. and M.A. contributed to the study concept and design. M.R. and A.C. led the acquisition of data. A.C. led the statistical analysis of data. M.R., A.C., M.W., S.A. and S.P.J. led the interpretation of data. M.R. and A.C. drafted the manuscript, and all authors critically revised the manuscript for important intellectual content. M.R. and S.P.J. obtained funding and provided supervision.

## Funding

## Competing interests

The authors declare no competing interests.
