## [Peer Review File · Nature Communications]

Inequalities in COVID-19 severe morbidity and mortality by country of birth in SwedenREVIEWER COMMENTS

Reviewer #1 (Remarks to the Author):

Prof Rostila and colleagues propose an analysis of the Swedish population registers focused on the increased risk of COVID-19 UCH and deaths in migrant groups. I'm always amazed by the quality and richness of the Swedish data, and I find the manuscript well-written, thorough and tackling an important issue. I am not completely satisfied by some aspects the statistical analysis. This manuscript still provides a coherent description of an important problem, and the conclusions are supported by the data. I just think that more could be done about this important question with this amazing dataset. I listed several points for the authors to consider, but after answering these I consider that this manuscript may be accepted for publication.

Major points

- The analysis is not perfect, but acceptable. In particular, I am not convinced that the chosen approach to examine the residual effect of the country of origin is to compare RRs from models with and without adjustment on SES, living conditions and vaccination status is entirely adequate. It is fine, and in line with the literature, but a more direct and reliable approach would have been to include interactions between these factors, and so be able to quantify the increase in risk due to the country of origin rather than indirectly comparing RRs with overlapping CIs. Of course this can become difficult with many variables, so techniques like dimensionality reduction or regularization might be necessary. The authors show several interactions in the supplementary but they are difficult to read, and the authors don't really make use of them.
- The approach is based on model comparison but no model selection tools are shown (eg AIC and the like). There is also no discussion of the potential overdispersion that may require using negative-binomial regression instead of Poisson.
- The authors also don't show any measure or plot of the goodness of fit.
- There is no mention of the incomplete ascertainment of COVID-19 related deaths, that has been shown in several countries (eg Riou et al, Nature communications 2023). Some of the patterns could be influenced by differential ascertainment. Can the authors elaborate on this?
- I'm a bit puzzled that nordics w/o Sweden still have a higher risk of ICH and death after adjusting for age, sex, SES, living conditions and vaccination status. I noticed that this category is older, so it may indicate that adjustment on age was not sufficient? Can the authors elaborate on this?
- The authors do not really conclude on an important result, which is that SES, living conditions and vaccination only partially explain the increased risk of UCH and deaths in migrants, especially during the early period. While it's difficult to explain the source of this residual risk, I think this should appear in the conclusions and abstract.

Minor points

- abstract, line 29: Just a detail, but I would avoid using the word "excess" here, I found it a bit confusing as the statement is not related to the concept of excess mortality.
- abstract, line 39: Also a small detail, but I would consider finding an alternative to "EU28/EEA" or "EU28" as I find it confusing with regards to the formal status of the UK and Switzerland.
- introduction, line 70: I assume "vaccinations have been decisive to reduce overall rates of severe illness and death"?
- methods, line 93: Please clarify the definition of country of origin. I assume it's country of birth of each individual? It's just that in some analyses, country of origin is defined as the country of

birth of the parents.

↓- methods, line 121: I assume ICH instead of ICU? Also, ICH care is improper as the C already means care.

- methods, line 122: Please clarify "with ensuing event recorded"

- Table 1: Please add confidence intervals for the rates.

- Table 2: Please add some explanation of the occupation SSYK codes. I also think that some of these variables (eg region) have more their place in the supplementary.

- All figures: the black colouring makes it difficult to see confidence intervals. The confidence intervals are also not defined in the caption. Last, the colouring does not correspond across figures, as black corresponds to M5 in figures 1 and 2 but to M4 in figure 3.

- results, line 185: typo "by for"

- Figure 2: I found figure 2 quite difficult to read. I would suggest colouring by wave and facetting by country, so it's easier to compare coefficients across waves. Not sure it's necessary to show both M1 and M5 here, M5 seems sufficient.

- results, line 211: While I understand what the authors mean, I disagree with the formulation that "any excess risk is fully explained by SES, living conditions and vaccination status". The fact that the CIs for M5 all include 1 does not imply that there is no effect, but rather that the data is compatible with no effect. It's the same mistake as concluding that "there is no difference" with a p-value > 0.05. Please reformulate. In addition, the CIs of RR with M1 and M5 largely overlap, so it is difficult to formally conclude to a difference with this indirect approach. A more adequate approach would have been to show interaction terms.

- results, line 228: I assume "the RR are are no longer significantly different from 1 when accounting for vaccination".

- Figure 3: If the focus is on RR reduction after adjusting by vaccination status, why not show the reduction directly?

- discussion, line 239: I don't think COVID-19 ever stopped from being considered a public health threat. Maybe it's not a public health priority anymore, but it's certainly still a threat.

- discussion, line 245: Another explanation for the decrease of disparities during the second wave and the rise during the third wave

- discussion, line 258: While the other explanations are convincing, I'm a bit skeptical of the explanation based on the vaccination campaign focused on older people first, aren't the models adjusted for age? This could be another indication that the age adjustment is not sufficient. Can the authors elaborate?

Reviewer #2 (Remarks to the Author):

This is an important study on inequalities in COVID-19 morbidity and mortality in Sweden in relation to country of birth. I have the following comments and suggestions to the authors:

* Major comments

1. Endpoints

a. ICU is a problematic endpoint as ICU treatment depends on many other factors than disease

severity, including availability of other treatments and the patient's chances of recovering after an ICU episode. There are therefore issues related to confounding from time period and selection bias, as also illustrated by Table 1, that should be discussed as a limitation. For this reason, I would suggest to include a composite endpoint, ICU or death, at least as a sensitivity analysis.

b. C19 hospitalization would have been an alternative endpoint, lack of such data from the National Patient Register in Sweden is also a limitation.

c. As the authors have data on c19 infections from SMINet, it is unclear why they were not used (mentioned as a limitation on row 303).

2. Conclusion regarding vaccinations

The authors claim that vaccination programs are effective in reducing c19 inequalities by country of birth. To back this claim up further, I would suggest to stratify analysis of wave 3 and 4 on vaccination status in order to estimate the country of birth-gradient in risk separate among vaccinated and unvaccinated. Previous work has shown how risk of severe c19 is related to comorbidities also among vaccinated (see eg. Euro Surveill 2022 Mar;27(9):2200121).

* Minor comments

1. Omicron, line 145

I would omit the word "possibly" as there are substantial evidence from other studies suggesting that Omicron is indeed milder.

2. Mortality risk, line 117

I would suggest to rephrase this sentence since "mortality risk" is a misnomer. Using Poisson regression, you are modelling c19 mortality rate and rate of c19 ICU care.

3. Table 1

These numbers are not so informative as the age distribution is clearly different across the groups. I would therefore suggest to add age-standardized rates.

4. Vaccinations, line 173 and Table 2

"Vaccination rates" is also a misnomer, I would suggest to use "vaccine uptake", "proportion vaccinated" or something similar.

Clarify date for the proportion vaccinated, is this by end of follow up whereas as all other data in Table 2 are at the end of 2019?

5. Third wave, line 219

I think there is a typo here, the sentence should refer to the second wave.

6. Discussion 255-257

As the increase in inequalities during wave 3 is seen also after adjustment for vaccinations there are likely to be other explanations than those put forward here. The suggested stratification of vaccination status could be used to explore this further.

7. Table 2

Table 2 is very long and therefore hard to read. I would therefore suggest to group some of the variables, e.g. regarding occupation and region.

Reviewer #3 (Remarks to the Author):

This is an original work that proposes to study inequalities in intensive care hospitalization and covid-19 mortality between Swedes and migrants. Other studies in Europe have shown excess mortality among migrants, but not with the social and living conditions information available in this

study at the individual level and not over the whole epidemic including the pre- and post-vaccination period, which is the case in this study.

The general methodology is clearly described, adequate and the conclusions are supported by the results and the quality of the available data.

I would like to make a few comments and requests for clarification concerning the data and their analysis in particular:

-Concerning the data, the level of exhaustiveness and the methods of collecting the registers and the cross-referencing carried out should be made explicit. The ability to make such cross-referencing is a strong point of this work that should be highlighted and better described in the interests of reproducibility and advocacy with other countries... Concerning the coding of variables, I wonder about the number of classes proposed for certain data and the possibility of grouping them together: e.g. for occupation, it would be relevant to use a classification that can be reproduced elsewhere by grouping certain classes together (International/European Standard Classification of Occupations); or the regions, which are in 25 classes and could be grouped together, for example, in relation to levels of incidence of infection.

-About the methodology: people with missing data should be better described, especially at the social level for the 1131 without country of birth, and at the social level and country of birth for the 13202 without DeSO neighbourhood. The definition of the 4 waves should be better described: how were the dates chosen? What source of information was used? When was vaccination widely introduced in Sweden?

-About the results: it seems to me that a major finding in the description of the data is the age difference between migrants and Swedes, and the fact that this age difference is associated with lower crude mortality rates but higher crude rates of intensive care hospitalization among migrants. This is not described in the results. It also seems to me that considering all migrants as the same group is questionable: there are differences in social and living conditions between North American, Asian and to a lesser extent European migrants.

-About the discussion, a paragraph dedicated to the strengths could be proposed. In addition, in the limitations, it seems to me that the fact that migrants can suffer discrimination which can influence access to care for example, but also be biologically embodied (via physiological stress for example) and influence biological functioning could be added as a potential mechanism between migrants and covid-19 related ICH and death.

-The conclusion could be developed in terms of the impact of these results for public action. Thus, regarding the last sentence of the conclusion, given the results observed, I am not sure that "equal" access to the vaccine is enough, but rather "equitable" access according to needs of the populations: we could then imagine priority access to the vaccine for migrant populations in the same way as was done for age. These results seem to me to favour an equitable approach (proportionate universalism) and not egalitarian in the sense of giving/proposing the same thing to everyone.

Minor points: put titles for figures 1 and 2 in appendix; Table 1: indicate the denominator for the mortality and incidence rates (per 100,000 people); Put the dates (at least original and end dates) on figures 1 and 2 of the appendix; Table 1 in the appendix: indicate in the title that it refers to ICH

REVIEWER COMMENTS

Reviewer #1 (Remarks to the Author):

Prof Rostila and colleagues propose an analysis of the Swedish population registers focused on the increased risk of COVID-19 UCH and deaths in migrant groups. I'm always amazed by the quality and richness of the Swedish data, and I find the manuscript well-written, thorough and tackling an important issue. I am not completely satisfied by some aspects the statistical analysis. This manuscript still provides a coherent description of an important problem, and the conclusions are supported by the data. I just think that more could be done about this important question with this amazing dataset. I listed several points for the authors to consider, but after answering these I consider that this manuscript may be accepted for publication.

RESPONSE: We thank the reviewer for these positive reflections.

Major points

- The analysis is not perfect, but acceptable. In particular, I am not convinced that the chosen approach to examine the residual effect of the country of origin is to compare RRs from models with and without adjustment on SES, living conditions and vaccination status is entirely adequate. It is fine, and in line with the literature, but a more direct and reliable approach would have been to include interactions between these factors, and so be able to quantify the increase in risk due to the country of origin rather than indirectly comparing RRs with overlapping CIs. Of course this can become difficult with many variables, so techniques like dimensionality reduction or regularization might be necessary. The authors show several interactions in the supplementary but they are difficult to read, and the authors don't really make use of them.

RESPONSE: We agree with the reviewer that the chosen modeling strategy is not perfect. Poisson regression models could involve problems in the study of infectious diseases since events are not independent. However, given the amount of data, the proposed solution of running Monte Carlo models is not feasible at this time. In this paper, the analysis is meant to be descriptive, almost as an extension of lifetables, where we are standardizing the rates by group distributions of age, sex, SES, living conditions and vaccine uptake. We hope to discern patterns of relative risks in the data and try to get an approximate picture of how covid-19 spread through the population, which migrant groups were the most vulnerable at which stages of the pandemic, and intuit some sense of explanatory factors for these relative risks. We do not think that the reduction in the incidence rate ratios when covariates relating to socioeconomic status or living arrangements numerically reflects an exact estimation but rather an indication of the relevance of such variables. Therefore, we are also cautious when making conclusions based on our results. Rather we add these factors and report a reduction in incidence rate ratios as a beginning into understanding the mechanisms for the observed higher standardized rates among groups of foreign born. We have added some discussion about this problem in the limitations section, see page 15, paragraph 1.

We also appreciate that Interactions between factors are especially important when considering migrant health since the mechanisms between socioeconomic status and health are moderated by migrant status. However, the interaction models become somewhat intractable with several dependent variables so in this analysis we have chosen to focus on the interaction with calendar

time (waves) and including risk factors that we think illustrate possible reasons for the differences in relative risks that are observed. We mention this in the limitations section, see page 15, paragraph 1.

- The approach is based on model comparison but no model selection tools are shown (eg AIC and the like). There is also no discussion of the potential overdispersion that may require using negative-binomial regression instead of Poisson.

RESPONSE: We now report AIC for all our models in the tables which is mentioned in the methods section, page 6. Interestingly, for covid-19 mortality it is the model which include the living conditions that has the lowest AIC. This suggests that adjusting for residence in elderly care homes seems to be the most important factor for covid-19 related deaths. We do also run least likelihood ratio tests to confirm that the interaction with time periods is significant which it is. This follows from the model assumptions as well, with the survival being constant for each time period, so splitting the follow up almost surely leads to a better estimation of the survival function.

We are aware of the problem of overdispersion in the Poisson models and ran a negative binomial model as well, as a sensitivity analysis. The results were qualitatively similar and the AIC was higher. Also, we do calculate robust standard errors to take into account the effects of heteroscedasticity and other model assumption such as overdispersion that do not hold.

- The authors also don't show any measure or plot of the goodness of fit.

RESPONSE: Given the large amount of data a plot of the goodness of fit is not very informative. However, the residual deviance is calculated for each model.

- There is no mention of the incomplete ascertainment of COVID-19 related deaths, that has been shown in several countries (eg Riou et al, Nature communications 2023). Some of the patterns could be influenced by differential ascertainment. Can the authors elaborate on this?

RESPONSE: We agree with the reviewer that incomplete ascertainment of COVID-19 related deaths could be a problem in our and other papers within this field of research. However, we do believe that Sweden generally has good quality of official statistics and registers. Although incomplete ascertainment might impact overall COVID-19 death rates, we believe that it would impact disparities by country of birth to a lesser extent. We have included the reference to Riou et al in our paper and added some discussion about this problem in the limitations section, see page 14, paragraph 1.

- I'm a bit puzzled that nordics w/o Sweden still have a higher risk of ICH and death after adjusting for age, sex, SES, living conditions and vaccination status. I noticed that this category is older, so it may indicate that adjustment on age was not sufficient? Can the authors elaborate on this?

RESPONSE: This result might be considered somewhat unexpected, but falls in line with what we know about the health of migrants from other Nordic countries to Sweden. Previous studies have

repeatedly demonstrated that Nordic migrants have a higher all-cause mortality risk and increased mortality from many specific causes (circulatory diseases, cancers, and various external causes) even after adjustment for age, sex and socioeconomic status:

Rostila, M., & Fritzell, J. (2014). Mortality differentials by immigrant groups in Sweden: the contribution of socioeconomic position. American Journal of Public Health, 104(4), 686-695.
<https://doi.org/10.2105/AJPH.2013.301613>

Wallace, M. (2022). Mortality Advantage Reversed: The Causes of Death Driving All-Cause Mortality Differentials Between Immigrants, the Descendants of Immigrants and Ancestral Natives in Sweden, 1997–2016. European journal of population, 38(5), 1213-1241.
<https://doi.org/10.1007/s10680-022-09637-0>

Wallace, M., & Wilson, B. (2022). Age variations and population over-coverage: Is low mortality among migrants merely a data artefact?. Population studies, 76(1), 81-98.
<https://doi.org/10.1080/00324728.2021.1877331>

Östergren, O., Korhonen, K., Gustafsson, N. K., & Martikainen, P. (2021). Home and away: mortality among Finnish-born migrants in Sweden compared to native Swedes and Finns residing in Finland. European journal of public health, 31(2), 321-325.
<https://doi.org/10.1093/eurpub/ckaa192>

- The authors do not really conclude on an important result, which is that SES, living conditions and vaccination only partially explain the increased risk of UCH and deaths in migrants, especially during the early period. While it's difficult to explain the source of this residual risk, I think this should appear in the conclusions and abstract.

RESPONSE: We agree with the reviewer that the fact that SES, living conditions and vaccinations only partially explained the increased risk of ICU admission and mortality. We should have highlighted this in the previous version of our manuscript. We have now highlighted this in the abstract where we have added that “In many migrant groups SES and living conditions contributed to the disparities, although elevated risks persist after adjustment for these factors” and discussion (page 12, paragraph 3).

Minor points

- abstract, line 29: Just a detail, but I would avoid using the word "excess" here, I found it a bit confusing as the statement is not related to the concept of excess mortality.

RESPONSE: We have rephrased this sentence to “Migrants have been more affected by the COVID-19 pandemic compared to the native host population”.

- abstract, line 39: Also a small detail, but I would consider finding an alternative to "EU28/EEA" or "EU28" as I find it confusing with regards to the formal status of the UK and Switzerland.

RESPONSE: We have now clarified that this group includes the UK and Switzerland, both in the abstract and in the methods section (page 4, paragraph 3).

- introduction, line 70: I assume "vaccinations have been decisive to reduce overall rates of severe illness and death"?

RESPONSE: Thanks for noticing this. We have amended the text accordingly (see page 3, paragraph 2).

- methods, line 93: Please clarify the definition of country of origin. I assume it's country of birth of each individual? It's just that in some analyses, country of origin is defined as the country of birth of the parents.

RESPONSE: We refer to the country of birth of the individual. We have rephrased and now use "country of birth" throughout the manuscript.

↓- methods, line 121: I assume ICH instead of ICU? Also, ICH care is improper as the C already means care.

RESPONSE: We agree the abbreviation used was somewhat misleading. We have now decided to use ICU admission throughout the manuscript.

- methods, line 122: Please clarify "with ensuing event recorded"

RESPONSE: We agree that this was not very well formulated. We have now rewritten part of the methods section and hope that the definition of the event and follow up is now clearer. It now reads as: "For mortality, follow-up started from March 1 2020 and ended on June 1 2021. On and between these two dates, individuals could experience the event (i.e., death from COVID-19) or be censored due to (a) death from a cause other than COVID-19 or (b) reaching the end of the follow-up period alive. For ICU admission, follow-up started from March 1 2020 and ended on June 1 2021. On and between these two dates, individuals could experience the event (i.e., admission to ICU) or be censored due to (a) death from COVID-19 (having not been admitted to the ICU), (b) death from a cause other than COVID-19, or through reaching the end of the follow-up period alive having not been admitted to the ICU". See page 5, paragraph 3 and page 1, paragraph 1.

- Table 1: Please add confidence intervals for the rates.

RESPONSE: Confidence intervals have now been added to the table.

- Table 2: Please add some explanation of the occupation SSK codes. I also think that some of these variables (eg region) have more their place in the supplementary.

RESPONSE: We have amended the occupation SSK which are the internationally comparable ISCO-08 codes and grouped according to standardized broad skill level (<https://ilostat.ilo.org/resources/concepts-and-definitions/classification-occupation/>). In addition, we have decreased the number of geographic residential regions to six larger regions. This is explained in the methods section see page 4, paragraph 3. We have also updated our results accordingly.

- All figures: the black colouring makes it difficult to see confidence intervals. The confidence intervals are also not defined in the caption. Last, the colouring does not correspond across figures, as black corresponds to M5 in figures 1 and 2 but to M4 in figure 3.

RESPONSE: We appreciate that the reviewer has made us aware of these shortcomings in the figures. The grey scale coloring has now been changed to npr color palette colors. The models 4 and 5 now have distinct colors from that palette. The confidence intervals are now mentioned in the caption as well.

- results, line 185: typo "by for"

RESPONSE: We have corrected the typo.

- Figure 2: I found figure 2 quite difficult to read. I would suggest colouring by wave and faceting by country, so it's easier to compare coefficients across waves. Not sure it's necessary to show both M1 and M5 here, M5 seems sufficient.

RESPONSE: We have now revised figure 2 so that it is faceted by country and colored by wave, and we only include M5. We hope that is now clearer how the disparities have changed over the course of the pandemic. Especially for ICU admission in the most vulnerable migrant groups, the disparities decreased over the waves such that no disparities remain in the last wave when adjusted for SES, living conditions and vaccination status.

- results, line 211: While I understand what the authors mean, I disagree with the formulation that "any excess risk is fully explained by SES, living conditions and vaccination status". The fact that the CIs for M5 all include 1 does not imply that there is no effect, but rather that the data is compatible with no effect. It's the same mistake as concluding that "there is no difference" with a p-value > 0.05. Please reformulate. In addition, the CIs of RR with M1 and M5 largely overlap, so it is difficult to formally conclude to a difference with this indirect approach. A more adequate approach would have been to show interaction terms.

RESPONSE: We agree with the reviewer and have reformulated this sentence to "However, as subsequent waves occurred, the relative risks gradually declined for most migrant groups and by the fourth wave no disparities in ICU admission rates were evident after adjusting for SES, living conditions, and vaccination status", see page 10, paragraph 1. Also see our previous response related to interaction terms.

- results, line 228: I assume "the RR are are no longer significantly different from 1 when accounting for vaccination".

RESPONSE: Yes, this is of course correct and we have changed the text accordingly. Apologies for the careless formulation.

- Figure 3: If the focus is on RR reduction after adjusting by vaccination status, why not show the

reduction directly?

RESPONSE: We chose to show the reduction by reporting the results from wave 4 since this is the time period in which everyone had been offered vaccinations.

- discussion, line 239: I don't think COVID-19 ever stopped from being considered a public health threat. Maybe it's not a public health priority anymore, but it's certainly still a threat.

RESPONSE: We agree with the reviewer and have rephrased to "...encompasses the entire pandemic to the point where COVID-19 was no longer considered a public health priority in Sweden", see page 11, paragraph 1.

- discussion, line 245: Another explanation for the decrease of disparities during the second wave and the rise during the third wave

RESPONSE: This was only the case for COVID-19 related mortality and not ICU admissions and as such any explanations are purely speculative. However, we propose an explanation which is a conjunction between a vaccination campaign which started with the very vulnerable, the older population and high-risk individuals, and a loosening of social restrictions which would increase exposure in vulnerable individuals in some migrant groups.

- discussion, line 258: While the other explanations are convincing, I'm a bit skeptical of the explanation based on the vaccination campaign focused on older people first, aren't the models adjusted for age? This could be another indication that the age adjustment is not sufficient. Can the authors elaborate?

RESPONSE: We agree with the reviewer that this explanation seems somewhat arbitrary when considering that the analyses are adjusted for age. However, we should add that the vaccinations did not only reach the older population first, vaccinations were also offered to risk groups (e.g. those with underlying diseases and health problems) and the vaccination coverage was also much lower in the older migrant population. Our explanation in the previous version of the manuscript was not clear enough. We believe that a combination of these factors could be a more nuanced explanation. We have rephrased and added some discussion, see page 12, paragraph 2:

"The introduction of vaccinations in wave 3 could contribute to increasing inequalities in COVID-19 related mortality between migrants and Swedish born. The vaccination campaign reached the older population and certain risk groups first (overrepresented by Swedish born) while covering the middle-aged migrant population to a lesser extent. Also, restrictions on movement and social interactions decreased as vaccinations were administered to the most vulnerable groups which could lead to higher rates of infections and community spread of the virus. These circumstances could have contributed to a lower protection in migrants who were then more exposed to the virus."

Reviewer #2 (Remarks to the Author):

This is an important study on inequalities in COVID-19 morbidity and mortality in Sweden in relation to country of birth. I have the following comments and suggestions to the authors:

* Major comments

1. Endpoints

a. ICU is a problematic endpoint as ICU treatment depends on many other factors than disease severity, including availability of other treatments and the patient's chances of recovering after an ICU episode. There are therefore issues related to confounding from time period and selection bias, as also illustrated by Table 1, that should be discussed as a limitation. For this reason, I would suggest to include a composite endpoint, ICU or death, at least as a sensitivity analysis.

RESPONSE: We agree with the reviewer that there might be confounding by time period and selection bias. We have now mentioned this problem in the limitation section, see page 14, paragraph 1. However, we do believe that this might be a less severe problem when it comes to disparities in ICU by country of birth since the Swedish public health care system should offer all citizens universal treatment (especially acute care).

Also we feel that the mechanisms for covid-19 related deaths and ICU admission are divergent to the point that a composite endpoint might not be so informative. Covid-19 related deaths were to a large degree a function of the spread of the virus in elderly care homes, while ICU admission better reflects the community spread of the virus. In addition, deaths are concentrated in the elderly while ICU admission affected younger cohorts. Since there is already an age effect of migrant groups being in general younger and healthier, we feel that creating a composite endpoint would not address the issues you so rightly point out.

b. C19 hospitalization would have been an alternative endpoint, lack of such data from the National Patient Register in Sweden is also a limitation.

RESPONSE: We agree. Unfortunately, we do not have access to this information in our data but hope to study hospitalization in the future.

c. As the authors have data on c19 infections from SMINet, it is unclear why they were not used (mentioned as a limitation on row 303).

RESPONSE: The data contains positive PCR tests responses only, and is therefore heavily biased toward those who were tested and do not accurately reflect covid-19 infections in the population. For example, hospital workers are overrepresented among those who were tested. Also, Sweden's testing capacity was very limited in the first wave as to be virtually non-existent. The difference between levels of positive tests and covid-19 IC and mortality is striking. Even when testing was widely available it was conducted or obtained by the individual themselves so what is observed is that more highly educated and higher earners have more positive tests since they take the tests to a much higher degree. Therefore, selection into testing behavior obscures any results obtained with this outcome. The authors have therefore chosen severe outcomes from covid-19 as these do not have the same problems.

2. Conclusion regarding vaccinations

The authors claim that vaccination programs are effective in reducing c19 inequalities by country of

birth. To back this claim up further, I would suggest to stratify analysis of wave 3 and 4 on vaccination status in order to estimate the country of birth-gradient in risk separate among vaccinated and unvaccinated. Previous work has shown how risk of severe c19 is related to comorbidities also among vaccinated (see eg. Euro Surveill 2022 Mar;27(9):2200121).

RESPONSE: Unfortunately, due to the small number of COVID-19 related ICU admissions and mortality per country/region of origin group in the unvaccinated the stratification by vaccination status in order to estimate country of birth gradient is not possible. However, since there is a socioeconomic gradient in vaccination uptake, it is probably the case that the unvaccinated have higher rates of co-morbidities that lead to more severe consequences of a COVID-19 infection and this would be very interesting to pursue with regards to country/region of birth. Perhaps if we obtain hospitalization data in the future we can further our analyses with associations between vaccination status, socioeconomic conditions, co-morbidities and possible moderation effects of country/region of birth. We mention the lack of data on co-morbidities in the limitations section, see 14, paragraph 1.

* Minor comments

1. Omicron, line 145

I would omit the word "possibly" as there are substantial evidence from other studies suggesting that Omicron is indeed milder.

RESPONSE: we have omitted "possibly" in this sentence.

2. Mortality risk, line 117

I would suggest to rephrase this sentence since "mortality risk" is a misnomer. Using Poisson regression, you are modelling c19 mortality rate and rate of c19 ICU care.

RESPONSE: Thank you for pointing this out. We are indeed modeling rates and estimating incidence rate ratios. This has now been clarified in the methods section, which reads as "We modeled mortality rates with confirmed COVID-19 or incidence rates of intensive care unit (ICU) admission with COVID-19 using Poisson regression analyses, and estimated incidence rate ratios dependent upon country/region of origin. See page 5, paragraph 3.

3. Table 1

These numbers are not so informative as the age distribution is clearly different across the groups. I would therefore suggest to add age-standardized rates.

RESPONSE: We have now added age and sex standardized mortality and incidence rates to Table 1.

4. Vaccinations, line 173 and Table 2

"Vaccination rates" is also a misnomer, I would suggest to use "vaccine uptake", "proportion vaccinated" or something similar.

Clarify date for the proportion vaccinated, is this by end of follow up whereas as all other data in

Table 2 are at the end of 2019?

RESPONSE: We now use “vaccination uptake” throughout the manuscript. We clarify in the methods that we split the data on date of first vaccination and that this is then a time-varying variable. In the descriptive table the proportions are calculated for the end of follow up which we also now mention in the results. Thanks for alerting us to these considerations.

5. Third wave, line 219

I think there is a typo here, the sentence should refer to the second wave.

RESPONSE: We have corrected the typo.

6. Discussion 255-257

As the increase in inequalities during wave 3 is seen also after adjustment for vaccinations there are likely to be other explanations than those put forward here. The suggested stratification of vaccination status could be used to explore this further.

RESPONSE: Thank you for this suggestion. Unfortunately, the stratified models are not possible given the small numbers for specific migrant groups. There are of course differential rates of comorbidities depending on country of origin and vaccination status, so other explanations are possible. One such possible explanation is that restrictions on movement and social interaction were gradually being lifted when the most vulnerable had been vaccinated leading to increased infection rates in the unvaccinated. However, the most likely explanation for the inequalities is a complex interaction between mitigating strategies leading to differential exposures and contact patterns, vulnerabilities in various social groups, and health care use (including vaccinations). Here we are only trying to highlight a few simplified relationships between country of birth, socioeconomic and living conditions, and vaccination status through the different phases of the pandemic. We cannot ascertain any one mechanism for the patterns observed for how the disparities evolved through the different phases of the pandemic. This is now discussed in the limitations section, see page 15, paragraph 1.

7. Table 2

Table 2 is very long and therefore hard to read. I would therefore suggest to group some of the variables, e.g. regarding occupation and region.

RESPONSE: : We have amended the occupation codes and grouped according to standardized broad skill level (<https://ilostat.ilo.org/resources/concepts-and-definitions/classification-occupation/>). In addition, we have decreased the number of geographic residential regions to six larger regions. This is explained in the methods section see page 4, paragraph 3 and page 5, paragraph 1. We have also updated our results accordingly. We hope the tables are more readable now.

Reviewer #3 (Remarks to the Author):

This is an original work that proposes to study inequalities in intensive care hospitalization and covid-19 mortality between Swedes and migrants. Other studies in Europe have shown excess mortality among migrants, but not with the social and living conditions information available in this

study at the individual level and not over the whole epidemic including the pre- and post-vaccination period, which is the case in this study.

The general methodology is clearly described, adequate and the conclusions are supported by the results and the quality of the available data.

I would like to make a few comments and requests for clarification concerning the data and their analysis in particular:

-Concerning the data, the level of exhaustiveness and the methods of collecting the registers and the cross-referencing carried out should be made explicit. The ability to make such cross-referencing is a strong point of this work that should be highlighted and better described in the interests of reproducibility and advocacy with other countries... Concerning the coding of variables, I wonder about the number of classes proposed for certain data and the possibility of grouping them together: e.g. for occupation, it would be relevant to use a classification that can be reproduced elsewhere by grouping certain classes together (International/European Standard Classification of Occupations); or the regions, which are in 25 classes and could be grouped together, for example, in relation to levels of incidence of infection.

RESPONSE: Concerning the coding of variables, we have amended the occupation codes and grouped according to standardized broad skill level (<https://ilostat.ilo.org/resources/concepts-and-definitions/classification-occupation/>). In addition, we have decreased the number of geographic residential regions to six larger regions. This is explained in the methods section see page 4, paragraph 3 and page 5, paragraph 1 and response to reviewer 2. We have also updated our results accordingly.

-About the methodology: people with missing data should be better described, especially at the social level for the 1131 without country of birth, and at the social level and country of birth for the 13202 without DeSO neighbourhood. The definition of the 4 waves should be better described: how were the dates chosen? What source of information was used? When was vaccination widely introduced in Sweden?

RESPONSE: We have now added some information on the social level of the individuals with missing data in the methods section with: "Of the 14416 individuals excluded, 48% were foreign born and 80% were in the lowest income quartile" page 5 paragraph 2.

We now also briefly described how the cut-off dates for the waves are found in the methods section: "The waves are illustrated in Figure 1 and 2 in the appendix for COVID-19 related IC and mortality, and were derived by finding when mortality cases per day goes under 10, and IC cases under 3. The cut-off date between wave 2 and 3 is when IC cases reaches a local minimum" page 7 paragraph 1.

We do mention in the methods section that vaccinations had begun on a large scale in the winter and spring of 2021, see page 6, paragraph 1.

-About the results: it seems to me that a major finding in the description of the data is the age difference between migrants and Swedes, and the fact that this age difference is associated with lower crude mortality rates but higher crude rates of intensive care hospitalization among migrants. This is not described in the results. It also seems to me that considering all migrants as the same group is questionable: there are differences in social and living conditions between North American, Asian and to a lesser extent European migrants.

RESPONSE: We now specify that the results are adjusted for age (and sex) in the text. We agree that there are major differences in social and living conditions between various groups based on region of origin. The description of Figure 1 at page 8 highlights some of these differences. We added a sentence that highlights these differences more clearly, see page 9 paragraph 2.

-About the discussion, a paragraph dedicated to the strengths could be proposed. In addition, in the limitations, it seems to me that the fact that migrants can suffer discrimination which can influence access to care for example, but also be biologically embodied (via physiological stress for example) and influence biological functioning could be added as a potential mechanism between migrants and covid-19 related ICH and death.

RESPONSE: We thank the reviewer for suggesting to highlight the strengths of our study. We have now added a paragraph briefly mentioning the strengths of our study/data. Although the Swedish health care system is public, we agree that migrants may suffer from discrimination that could influence their access to care/proper forms of treatment. We have now included a discussion of such a possibility, see page 12 paragraph 2.

-The conclusion could be developed in terms of the impact of these results for public action. Thus, regarding the last sentence of the conclusion, given the results observed, I am not sure that "equal" access to the vaccine is enough, but rather "equitable" access according to needs of the populations: we could then imagine priority access to the vaccine for migrant populations in the same way as was done for age. These results seem to me to favour an equitable approach (proportionate universalism) and not egalitarian in the sense of giving/proposing the same thing to everyone.

RESPONSE: We agree with the reviewer and have revised the conclusions of the study accordingly, see page 15, paragraph 2. "While preventive policies should focus on the social and living conditions of migrants in the absence of a vaccine, supporting equitable access according to the needs of the migrant population is the most important preventive strategy in order to mitigate disparities on COVID-19 related health outcomes. Also see revised conclusion in the abstract.

Minor points: put titles for figures 1 and 2 in appendix; Table 1: indicate the denominator for the mortality and incidence rates (per 100,000 people); Put the dates (at least original and end dates) on figures 1 and 2 of the appendix; Table 1 in the appendix: indicate in the title that it refers to ICH

RESPONSE: We have now added titles and dates in figure 1 and 2 in the appendix. We have also indicated that incidence rates are per 100, 000 people in table 1. We have also specified ICU admission in table 1 in the appendix.

REVIEWER COMMENTS

Reviewer #1 (Remarks to the Author):

The authors thoroughly revised their manuscript and answered all of my questions. I consider that this manuscript is now suitable for publication.

Reviewer #2 (Remarks to the Author):

The authors have responded to all of my comments of the previous version and incorporated appropriate changes. I only have one major and a few minor comments left:

Major comment:

1. The results presented in Figure 3 are the basis for much of the conclusions regarding the inequalities across the waves. To me it is not clear why the authors chose to present results from a fully adjusted model, including also variables such as vaccination status that are on the causal pathway (and thus not confounding factors in a strict sense). The authors should motivate this choice of model in relation to the aim of the study. In the Appendix, it would also be interesting to see the RR-estimates associated with country of origin under model 1-5 for the different waves 1-4. This would be a useful addition to Appendix Table 3 that is a bit hard to follow.

Minor comments:

1. Appendix Table 2 - Subheading for age category is missing in the table body.
2. Appendix Table 4 - State that the table contains RRs.

Reviewer #3 (Remarks to the Author):

The authors have responded to all my comments

RESPONSE LETTER

Reviewer #1 (Remarks to the Author):

The authors thoroughly revised their manuscript and answered all of my questions. I consider that this manuscript is now suitable for publication.

Reviewer #2 (Remarks to the Author):

The authors have responded to all of my comments of the previous version and incorporated appropriate changes. I only have one major and a few minor comments left:

Major comment:

1. The results presented in Figure 3 are the basis for much of the conclusions regarding the inequalities across the waves. To me it is not clear why the authors chose to present results from a fully adjusted model, including also variables such as vaccination status that are on the causal pathway (and thus not confounding factors in a strict sense). The authors should motivate this choice of model in relation to the aim of the study. In the Appendix, it would also be interesting to see the RR-estimates associated with country of origin under model 1-5 for the different waves 1-4. This would be a useful addition to Appendix Table 3 that is a bit hard to follow.

RESPONSE: We agree that only showing results from the fully adjusted model did not fully support our conclusions. Therefore, we added the results from the models adjusted for age and sex for the waves in the top panels in the figure, and hope the figure is still legible. We also added the RR and CI that are used in the figures (for M1, and M4, and M5) in tables 4 and 5 in the appendix. We hope that the details of the models are clearer and easier to follow now.

Minor comments:

1. Appendix Table 2 - Subheading for age category is missing in the table body.

RESPONSE: Thank you for alerting us to this, a part of the table had been cut when editing. The subheading and the missing values are now in the table.

2. Appendix Table 4 - State that the table contains RRs.

RESPONSE: Apologies for the oversight in the table headings. They have now been amended to: "Association between country/region of origin and Covid-19 related ICU admission / mortality for the seven time periods: RRs and 95% CIs for M1, M4, and M5 model specifications"

Reviewer #3 (Remarks to the Author):

The authors have responded to all my comments